# The Impact of a Revised National Childhood Immunization Schedule on Vaccination Defaulters

**DOI:** 10.3390/vaccines11040859

**Published:** 2023-04-17

**Authors:** Ngiap Chuan Tan, Jeremy Pang, Eileen Koh

**Affiliations:** 1SingHealth Polyclinics, Singapore 150167, Singapore; tan.ngiap.chuan@singhealth.com.sg; 2SingHealth Duke-NUS Family Medicine Academic Clinical Programme, Singapore 150167, Singapore; 3Duke-NUS Medical School, Singapore 169857, Singapore; e0036882@u.nus.edu

**Keywords:** childhood immunization, catch-up vaccinations, combination vaccines

## Abstract

Immunization schedules affect community vaccine uptake rates, especially in children who have defaulted on their regular immunization timelines. In 2020, Singapore revised its National Childhood Immunization Schedule (NCIS) to incorporate two new combination vaccines: the hexavalent hepatitis, diphtheria, acellular pertussis, tetanus (DTaP), hemophilus influenzae b (Hib), inactivated poliovirus (IPV) (6-in-1), and the quadrivalent measles, mumps, rubella, and varicella (MMRV) vaccines, thus reducing the mean number of clinic visits and vaccine doses by two. Our database study aims to evaluate the impact of the 2020 NCIS on catch-up vaccination uptake rates in children at 18 and 24 months of age and the catch-up immunization rates of individual vaccines at two years. Vaccination data from two cohorts, in 2018 (n = 11,371) and in 2019 (n = 11,719), were extracted from the Electronic Medical Records. Catch-up vaccination rates increased by 5.2% and 2.6% in children on the new NCIS at 18 and 24 months, respectively. The uptake of individual 5-in-1 (DTaP, IPV, Hib), MMR, and pneumococcal vaccines increased by 3.7%, 4.1%, and 1.9% at 18 months. Reduced vaccination doses and visits in the new NCIS bring direct and indirect benefits to parents and promote vaccination adherence for their children. These findings highlight the importance of timelines in improving catch-up vaccination rates in any NCIS.

## 1. Introduction

Childhood vaccinations remain cost-effective, preventive health measures globally [1,2]. Such vaccinations decrease the morbidity and mortality of common pediatric communicable diseases and minimize national economic and healthcare burdens [3,4]. Childhood vaccinations against diseases such as tetanus, pertussis, diphtheria, poliomyelitis, hepatitis B, and hemophilus influenzae type B have helped to prevent over 2–3 million deaths yearly [5], saving approximately 750,000 children from disability [1]. Conversely, intentional refusal or delays in immunization have led to disruptions in local vaccination coverage, reduced overall herd immunity, and subsequently, increased risk of disease outbreaks [6]. Immunization delays have been linked to the recent 2008 measles outbreaks in the US [7,8]; defaulting vaccinations also warrants re-administration, which increases the risk of developing adverse reactions [9].

Vaccination coverage among children relies heavily on the healthcare system to provide timely, accessible, and affordable health services. Apart from direct health detriments to unvaccinated children, parents bear indirect costs when they fail to adhere to timely vaccination schedules. These indirect costs relate to lost work time, productivity losses associated with caregiving roles, time spent traveling to visit healthcare providers, and the rescheduling of vaccinations [10,11]. Hospitalizations due to severe pediatric infections burden the parents with additional direct and indirect costs [12]. From a health systems perspective, there is also an increased strain on healthcare infrastructure and resources which must be allocated for the treatment of vaccine-preventable diseases.

Singapore is a developed island state with a comprehensive childhood vaccination program. In 2019, the percentages of children at 1 year of age who were vaccinated for diphtheria, tetanus, and pertussis (DTaP), measles, and hepatitis were 97%, 95%, and 96%, respectively [13], edging out the Organization for Economic Co-operation and Development (OECD) coverages of 95%, 95%, and 91% for DTaP, measles, and hepatitis vaccines, respectively [13]. The local immunization of infants and preschool children is primarily covered by three main healthcare clusters of public primary care clinics (polyclinics) and private general practitioners. 

Timely vaccinations are essential to protect children from related infections during their vulnerable period. The timeliness of childhood vaccinations is executed in accordance with the National Childhood Immunization Schedule (NCIS) [14]. Prior to the recent NCIS revision, infants up to the age of 2 were scheduled to receive 13 vaccine doses covering hepatitis B (HepB), diphtheria, tetanus, acellular pertussis, inactivated polio vaccine, haemophilus influenza b (5-in-1), measles, mumps, rubella (MMR), and pneumococcal (PCV13) over the course of eight visits [4]. Only the measles and diphtheria vaccinations are compulsory by law [4,15].

Under the revised NCIS 2020 schedule, the introduction of the hexavalent (6-in-1) hepatitis B, tetanus, acellular pertussis, diphtheria, hemophilus influenzae type B, and inactivated polio vaccine along with the quadrivalent mumps, measles, rubella, and varicella (MMRV) vaccine reduces the overall vaccination doses to eleven (from thirteen) over six visits (from eight) by two years of age. The introduction of combination vaccines is postulated to reduce the overall vaccination cost to parents and healthcare providers [1,16], reduce anxiety over multiple injections [17], and improve the compliance and timeliness of vaccinations [18]. 

A previously published study demonstrated that 1st-year vaccination uptakes rates were increased by 2.1% in participants on the new 2020 NCIS [19]. This however falls short of nationally recorded coverages of individual vaccines [4] and underpins the need to pursue catch-up vaccinations after the 1st year of age, thus ensuring that children in Singapore are able to meet their vaccination milestones.

This study aims to compare the overall uptake rate of catch-up vaccinations in children by two years of age, before and after the implementation of the new 2020 NCIS. The secondary objective is to determine the catch-up immunization rate of each vaccine and its trend over 2 years. The findings allow the identification of gaps and the designing of interventions to address the barriers for timely vaccination [20]. 

## 2. Materials and Methods

### 2.1. Study Sites and Population

This retrospective database study reviewed the vaccination records of 23,090 children aged 2 years old at the time of data collection. These children received their childhood vaccinations from one of three clusters of public care clinics (polyclinics), comprising eight such polyclinics located in the eastern region of Singapore. The operational nature of Singapore’s healthcare system allows the public to seek follow-up consultations with their preferred primary care physician, at either SingHealth polyclinics or private general practitioner (GP) clinics [21,22]. 

The study population consisted of children born between 1 January 2018 and 31 December 2019, categorized into 2018 and 2019 cohorts with 11,371 and 11,719 children, respectively. Children in each cohort were further subdivided by gender. 

Vaccination records of each study cohort were extracted from the Electronic Medical Records (EMR) database at two study time points: (1) between 1 July 2019 and 30 June 2021 to determine the initial 18-month vaccination uptake, and (2) between 1 November 2020 and 31 December 2021 to determine the final 24-month vaccination catch-up uptake rates. 

### 2.2. Study Design

The primary outcomes of this study are the initial 18-month and 24-month total catch-up vaccination uptake rates between both the 2018 and 2019 cohorts. Children in the 2018 cohort underwent the old 2016 NCIS vaccination protocol (Table 1). Children in the 2019 cohort transited to the new NCIS vaccination protocol in 2020 (Table 2), allowing us to evaluate if the introduction of combination vaccines resulted in higher vaccination rates.

To further streamline our dataset, children who had only received 1 dose of any vaccine (Hep B, 5-in-1, MMR, PCV13) were excluded from data collection. This effectively excluded participants who only visited SingHealth Polyclinics once and subsequently defaulted on their future follow-up vaccination appointments. 

As both the 2018 and 2019 cohorts were born before the new NCIS revision in 2020, the primary criterion for vaccination uptake was adjusted accordingly. The vaccines under study are: two hepatitis B vaccines, four 5-in-1 vaccines, two MMR vaccines, and three PCV13 vaccines by 24 months of age. These vaccines were covered under the old NCIS guidelines in 2016, in which influenza and varicella vaccinations were yet to be introduced [4]. The baseline vaccine uptake was determined at 18 months, according to the number of fulfilled vaccination doses. The catch-up vaccination uptake was assessed 6 months later, at 24 months of age, comparing vaccine uptake in children who followed the old (2016) and new (2020) NCIS protocols. 

### 2.3. Vaccination Schedule

On 1st November 2020, the MOH rolled out the new National Childhood Immunization Schedule (NCIS) that included two new combination vaccines: the hexavalent hepatitis B, diphtheria, acellular pertussis, tetanus, hemophilus influenzae type b, inactivated polio vaccine (6-in-1) and the quadrivalent measles, mumps, rubella, varicella (MMRV) vaccine. Notably, the new 2020 NCIS incorporates both Varicella and seasonal influenza vaccines that were not recommended in the previous iteration. The usage of both the hexavalent 6-in-1 and the quadrivalent MMRV combination vaccines has reduced total vaccination doses to 11 (from 13) over 6 visits (from 8) by two years of age. 

### 2.4. Data Extraction, Processing, and Audit

All clinical data, including vaccination history and consultation notes, are recorded in the Electronic Medical Records (EMR) following input into the Sunrise Clinical Manager (SCM) platform. The collected data are then stored within a single enterprise data repository, termed the Electronic Health Intelligence System (eHints). eHints serves as SingHealth’s enterprise analytic platform that integrates information from multiple healthcare transactional systems to provide healthcare staff with high-quality clinical patient data [23]. The SHP Research Informatics team subsequently extracted the vaccination data of children from birth to 24 months of age from the eHints database. 

### 2.5. Statistical Analysis

Statistical analysis was performed using IBM SPSS version 27.0. Categorical variables were represented as means and standard deviations (SD). Chi-squared tests were used to compare categorical differences between vaccination uptake rates in both study cohorts. We predetermined a *p*-value of <0.05 as statistical significance, and a two-tailed test was performed for all analyses.

## 3. Results

At the 18-month time point in each cohort, the vaccination uptake rate increased by 5% between the old and new NCIS cohorts (Table 3). At 24 months, this difference was still statistically significant (*p*-value < 0.001), with a 2.6% increase in overall catch-up vaccination uptake in the new NCIS cohort. The catch-up vaccination uptake at 24 months was 63.4% and 66.0% for the old and new NCIS cohorts, respectively. Among 8142 participants who did not meet the catch-up uptake criteria at 24 months, 48.9% belonged to the new NCIS cohort, while 51.1% belonged to the old NCIS cohort. 

The results showed a significant increase in the 18-month 5-in-1 vaccine uptake rate of 3.7%, from 46.5% to 50.2% in the new NCIS cohort, *p*-value < 0.001. A higher 33.5% of the 2018 cohort required catch-up with the 5-in-1 vaccination at 18 months, while just 29.8% of the 2019 cohort required catch-up of the 5-in-1 vaccination. Looking further at MMR catch-up vaccination uptake rates, 76.8% of children in the 2018 cohort were fully vaccinated (≥2 doses) at 18 months, compared to 80.9% of participants in the 2019 cohort, reflecting a 4.1% increased uptake, *p*-value < 0.001 (Table 3). However, MMR vaccine uptake at 24 months was lower in the 2019 cohort at 88.3%, compared to 89.9% in the 2018 cohort. 

PCV13 vaccination uptake rates registered a 1.9% and 2.1% increase at both the 18- and 24-month catch-up time points in the new NCIS cohort, *p*-value < 0.001. Similarly, the hepatitis B vaccine uptake increased by 0.9% at 24 months, *p* = 0.058. 

Of the 15,308 of children, 50.8% who completed the vaccinations were males, showing no significant gender difference in the uptake (Table 4). There was also no statistically significant difference in individual vaccine uptake among both male and female genders.

## 4. Discussion

The overall catch-up vaccination uptake rates are comparable for children enrolled in the new NCIS, with even a small increase of 2.6% by 24 months (Table 1). The most significant increase in vaccination uptake rates is observed at 18 months in children receiving the 5-in-1 vaccination, with a 3.7% increase in the 2019 cohort. Implementing a 6-in-1 combination vaccine that incorporates both hepatitis B and 5-in-1 antigens likely accounts for this increase in vaccination coverage. A reduction in two overall vaccination visits translates to decreased costs to parents [17] and bolsters catch-up vaccination uptake rates in the 2020 NCIS cohort. From an organizational standpoint, the conveniences afforded from reduced vaccination visits reduces the manpower utilization and burden on healthcare services. Freed up resources and manpower can be channeled into other essential primary care services.

Despite a small increase, the overall PCV13 vaccination uptake at 24 months under the new NCIS registers at 73.2% (Table 3), well below the national target of 95%. Nationally, pneumococcal vaccine coverage is also one of the lowest at just 81.5% in 2017 [4], based on the completion of three doses by 2 years of age. Several reasons are cited for this suboptimal coverage: knowledge deficiency on the susceptibility to pneumococcal disease and a lack of perceived threat and perceived benefits to children [24]. Ironically, childhood vaccinations have fallen victim to their own success, as the effectiveness of current programs lowers the perceived risks of vaccine-preventable diseases. Before the NCIS revision in 2020, PCV13 posed a financial burden to some families and potentially increased barriers to its uptake. Given the full subsidization of PCV13 in 2020 [25], it remains unclear if public education, the increased communication of benefits from healthcare providers, and parental decision support could lead to improved vaccine uptake. 

Despite the convenience, safety, and non-inferiority of combination vaccines, it is worth noting that overall vaccination uptake rates fall short of the national recommended target at 24 months of age, with just 63.4% and 66.0% completion in the 2018 and 2019 cohorts, respectively. Firstly, vaccination data from private healthcare providers such as the general practitioners (GPs) and pediatricians in both public and private hospitals were not captured in this study, and this omission likely results in an underestimation of total catch-up vaccination rates. GPs operate longer and more flexible clinic hours, with increased accessibility and perceived personal care [13,26]. Parents who miss the immunization schedule for their children at polyclinics can opt for catch-up vaccination at GP clinics. Furthermore, the data were derived from 2018 to 2021 EMR records, coinciding with the COVID-19 pandemic. A partial lockdown termed a “circuit breaker” was imposed in Singapore from April to June 2020 [27]. While essential services such as childhood vaccinations remained available during this period, polyclinics nevertheless reported lower attendances for all vaccination services [28]. Zhong et al. attributed this decline to increased parental hesitancy and fear of contracting the contagion within healthcare facilities [28]. Such psychological factors may affect the childhood vaccination uptake rate at polyclinics.

Another critical barrier to catch-up vaccination rates is the rescheduling of missed appointments by parents. Parents change appointments for their children’s vaccinations for various reasons such as the latter’s acute illnesses or conflicts of timing with their social or professional work [29]. Rescheduling can be carried out using an online health portal such as the “Health Buddy” mobile application, in-person via an on-site e-kiosk at a polyclinic, or via a direct phone call [30]. Paradoxically, this perceived ease of rescheduling via multiple modalities could also perpetuate poor adherence to vaccination. Such defaults result in care delivery lapses and additional costs and work burdens for nurses and clinic administration. The unvaccinated child is also at risk of infections. An innovative chatbot to raise the vaccine literacy of the parents, automate the rescheduling of vaccination appointments, serve reminders, and screen the health status of the child is in the pipeline. 

Combination vaccines such as hexavalent TDaP-HiB-HepB-polio and quadrivalent MMRV have been extensively documented to be well tolerated, with a comparable safety profile to the administration of individual vaccines [31,32,33,34,35,36]. The most common adverse events reported in MMRV vaccinations were injection site reactions, an allergic rash, and fever, but they were not significantly different from individual vaccine counterparts [31]. Likewise, immunogenicity profiles elucidated a similar seroconversion rate in MMRV vaccine groups, with the measles component reporting a higher geometric mean titer ratio than MMR+V/MMR groups [32]. Hexavalent TDaP-HiB-HepB-polio vaccines also demonstrated non-inferiority to pentavalent Infanrix hexa^TM^ in regard to seroprotection and vaccine response rates for all component antigens following primary vaccination [37,38]. Despite the similar safety profiles and immunogenicity of combination vaccinations, perceived vaccine safety and effectiveness are still cited by parents as important contributory factors to vaccine uptake hesitancy [39]. 

One proposed intervention to address the parental perception of childhood vaccine safety is the implementation of active surveillance measures to track adverse events following immunizations (AEFIs). Active surveillance has the propensity to increase the detection of AEFIs compared to traditional post-licensure passive surveillance [40], with the median interval from vaccination to symptom onset being approximately 6 days [41]. A recent trial in Switzerland deployed an SMS-based surveillance system and recorded high overall response rates, a short time-to-respond, and highly favorable acceptance rates among participants [42]. The resultant increase in AEFI reporting raises the awareness of the safety profile of childhood combination vaccines, reduces mistrust regarding vaccine safety [40], augments the parental perception of combination vaccination efficacy [39], and ultimately strengthens public confidence in national immunization programs [40,41]. 

Vaccine hesitancy is complex, multifaceted, and deeply influenced by individual motivations, health-seeking behaviors, underlying assumptions, and personal heuristics that impact perception of risks [43]. Vaccine hesitancy can be mitigated by supporting individual decision making based on transparent and objective, evidence-based information [44]. Improving vaccine literacy among parents remains a key pillar in modifying their decision patterns to vaccinate their children. Parental concerns and beliefs over childhood vaccinations may fluctuate over time [45], necessitating the need for personalized health messaging together with broad-stroke nationwide educational campaigns. Personalized vaccination messages can also help tackle the misinformation, driven largely by social media platforms [44], that runs rampant during the adoption of national vaccination policies, as evidenced by recent COVID-19 initiatives. Furthermore, vaccination content should be tailored to the hesitancy level of the receiver. According to Olsen et al., parents who are mildly cautious appreciate information regarding the childhood vaccination benefits or risks, whereas highly cautious parents prefer information regarding the rationale for combination vaccination use and technical information [46]. Decision-support aids have been developed to support parents who are ambivalent in vaccinating their children. For ease of access and with the ubiquitous use of smart phones among parents, digitalized, voice-over, or animated parental decision-support aids have shown preliminary efficacy in increasing influenza vaccine uptake by enhancing parental decision-making processes and the provision of information on both the safety and risks of the vaccine [47]. The content of such aids can potentially be extended to the other childhood vaccines. 

Fear and anxiety can significantly affect both parents and children during their immunization visits, which may hinder their subsequent vaccine uptake [48,49]. The usage of needles during the immunization process generates fear, distress, and the perception of pain in children [49]. The introduction of combination childhood vaccines partially mitigates anxiety by decreasing the vaccination frequencies. However, procedural pain cannot be addressed by changes in vaccination scheduling alone. Current research into the use of immersive virtual reality (VR) technologies shows promise in attenuating the pain associated with immunization procedures [50]. Children’s fear scores along with parental anxiety scores were significantly decreased in the VR intervention group. The incorporation of VR technologies, together with a simplified immunization schedule, can positively influence a child’s immunization journey and enhance their adherence to their immunization schedule.

Time constraints, missed vaccination opportunities, costs, and safety concerns about multiple vaccinations have long been cited as barriers to vaccine uptake [43,51]. Addressing these barriers and the introduction of combination vaccines and a simplified NCIS are strategies to help parents to catch up with vaccinations. Moving forward, digital decision-support aids should help parents to raise their vaccine literacy and reduce their hesitancy with vaccinating their children. Other measures include the establishment of an active surveillance system to better inform parents on vaccine safety and risk profiles and further minimize their hesitancy. An automated appointment system which will facilitate catch-up vaccine scheduling is in the pipeline. Immersing children in virtual reality (VR) during their vaccination will improve their experience during their clinic visit and promote their timely completion of their entire immunization schedule. 

## 5. Strength and Limitations

This is likely the first large database study that examines the catch-up vaccination rates after the new NCIS was implemented. Furthermore, its implementation comes amidst the backdrop of a global pandemic in which childhood vaccination rates experienced their most significant sustained decline and highest default rate [52]. Nonetheless, the favorable data and up-trending vaccination coverage highlights the need for a periodic review of the NCIS, affords foresight into potential challenges, and lays the foundation for future public health research methodologies in the same field.

Several limitations have been identified in this study. The number of parents who have defaulted their follow-up vaccinations with SHP in favor of private family physicians or other healthcare clusters remains unclear as such information is not recorded in the institution EMR. Data such as reasons for abscondment of any vaccination, irrespective of healthcare provider, were also not available in this study. It is hence difficult to ascertain if parents are adhering to recommended catch-up intervals, and if there were extenuating circumstances to warrant their default. Such information could provide greater insight into the health-seeking behaviors of parents, allowing the identification of additional barriers to vaccination hesitancy not addressed by the new 2020 NCIS. 

## 6. Conclusions

The implementation of the new 2020 NCIS has improved the catch-up vaccination uptake rate in children at both 18 and 24 months of age, highlighting the role of periodically reviewing national immunization schedules in optimizing childhood vaccination adherence. The usage of combination vaccines translates to a decreased total number of primary care visits, healthcare system utilization, and productivity loss from rescheduled vaccination appointments. Enhancements in overall catch-up vaccine coverage reduce childhood mortality and disease burden, ultimately preventing costly hospitalizations and disabilities among the pediatric population.

## Figures and Tables

**Table 1 vaccines-11-00859-t001:** 2016 National Childhood Immunization Schedule (NCIS) Singapore.

2016 NCIS.									
	Birth	1 Months	3 Months	4 Months	5 Months	6 Months	12 Months	15 Months	18 Months
BCG	D1								
Hepatitis B	D1	D2				D3			
Diphtheria, acellular pertussis, tetanus			D1	D2	D3				B1
Inactivated poliovirus			D1	D2	D3				B1
Hemophilus influenzae type b			D1	D2	D3				B1
Pneumococcal conjugate			D1		D1		B1		
Measles, mumps, rubella							D1	D2	

**Table 2 vaccines-11-00859-t002:** New 2020 National Childhood Immunization Schedule (NCIS) Singapore.

2020 NCIS.							
	Birth	2 Months	4 Months	6 Months	12 Months	15 Months	18 Months
BCG	D1						
Hepatitis B	D1	D2		D3			
Diphtheria, acellular pertussis, tetanus		D1	D2	D3			B1
Inactivated poliovirus		D1	D2	D3			B1
Hemophilus influenzae type b		D1	D2	D3			B1
Pneumococcal conjugate			D1	D2	B1		
Measles, mumps, rubella					D1	D2	
Varicella					D1	D2	

**Table 3 vaccines-11-00859-t003:** Rate of vaccination uptake between 2018 and 2019 cohorts at both 18-month and 24-month time points; ^1^ 5-in-1 refers to combined diphtheria, tetanus, acellular pertussis, inactivated poliovirus, and hemophilus influenzae type b vaccine.

	2018 COHORT 18M, *N* (%)	2019 COHORT 18M, *N* (%)	*p*-Value	2018 COHORT CATCH-UP 24M, *N* (%)	2019 COHORT CATCH-UP 24M, *N* (%)	*p*-Value
**ALL VACCINES**						
**UNFULFILLED**	7160 (63.0)	6768 (57.8)		4159 (36.6)	3983 (34.0)	
**FULFILLED**	4211 (37.0)	4951 (42.2)	<0.001	7212 (63.4)	7736 (66.0)	<0.001
**5-IN-1 ^1^**						
**≤** **3**	6084 (53.5)	5839 (49.8)		2218 (19.5)	2423 (20.7)	
**≥** **4**	5287 (46.5)	5880 (50.2)	<0.001	9153 (80.5)	9296 (79.3)	0.027
**HEPATITIS B**						
**≤** **1**	1912 (16.8)	1864 (15.9)		1869 (16.4)	1819 (15.5)	
**≥** **2**	9459 (83.2)	9855 (84.1)	0.062	9502 (83.6)	9900 (84.5)	0.058
**MMR**						
**≤** **1**	2636 (23.2)	2242 (19.1)		1151 (10.1)	1368 (11.7)	
**≥** **2**	8735 (76.8)	9477 (80.9)	<0.001	10,220 (89.9)	10,351 (88.3)	<0.001
**PCV13**						
**≤** **2**	3352 (29.5)	3235 (27.6)		3286 (28.9)	3140 (26.8)	
**≥** **3**	8019 (70.5)	8484 (72.4)	0.002	8085 (71.1)	8579 (73.2)	<0.001

**Table 4 vaccines-11-00859-t004:** Vaccination uptake rates between male and female participants for each individual vaccine.

	FEMALE, *N* (%)	MALE, *N* (%)	*p*-Value
**ALL VACCINES**			
**UNFULFILLED**	3931 (48.3)	4211 (51.7)	
**FULFILLED**	7358 (49.2)	7590 (50.8)	0.171
**5-IN-1**			
**≤3**	2226 (48)	2415 (52)	
**≥4**	9063 (49.1)	9386 (50.9)	0.157
**HEPATITIS B**			
**≤1**	1762 (47.8)	1926 (52.2)	
**≥2**	9527 (49.1)	9875 (50.9)	0.140
**MMR**			
**≤1**	1194 (47.4)	1325 (52.6)	
**≥2**	10,095 (49.1)	10,476 (50.9)	0.113
**PCV13**			
**≤2**	3092 (48.1)	3334 (51.9)	
**≥3**	8197 (49.2)	8467 (50.8)	0.144

## Data Availability

Data available on request due to restrictions.

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
