# Peer review of "The Impact of a Revised National Childhood Immunization Schedule on Vaccination Defaulters"

_vaccines, 2023, doi:10.3390/vaccines11040859_

Round 1

Reviewer 1 Report

In this manuscript, the authors aims to evaluate the impact of the 2020 NCIS on catch-up vaccination uptake rates in children at 18 and 24 months of age, and the catch-up immunization rates of individual vaccines at two years. The logic of this study is clear and the results are simple-to-understand. However, it is better to dig deeper for the study. For instance, the authors are suggested to survey and summarize the real reasons of incompleted vaccination uptake with a table. Additionally, the authors are suggested to add some discussions about how to make the population more positive to the vaccination. One possible solution is to optimize or customize the individual vaccination plan by dynamic monitoring of vaccination effectiveness. Some related critical papers could be referred to, such as Ilies Benotmane et al., Journal of Personalized Medicine 2022, 12(7), 1107; Mariana C. Castells et al., New England Journal of Medicine 2021, 384:643-649; Haoyang Tong et al., Biosensors & Bioelectronics 2022, 114449. Therefore, a major revision is suggested before its publication in Vaccines.

Author Response

Results

  1. Survey and summarize real reasons of incomplete vaccination uptake with a table

Thank you for the suggestion. As our retrospective database study involved the extraction of data from the electronic health records repository, we were not able to capture qualitative data on the exact reasons for vaccination default. We are only able to quote reasons for vaccination hesitancy/default based on recent literature reviews. However, we acknowledge the importance of understanding the reasons for the incomplete vaccination uptake, which should be assessed in a future study with the appropriate study design. It has been included as one of the limitations of our current study.

Discussion

  1. Authors are suggested to add some discussions about how to make the population more positive to the vaccination. Either through customizing individual vaccination plans or monitoring of vaccine effectiveness

We appreciate the suggestion to discuss individual vaccination plans as a recommendation to improve overall vaccination uptake. Therefore, we have included a discussion point to explore the use of SMS-based active surveillance system to report combination vaccine safety and adverse outcomes in lay language, as a potential measure to allay parental concerns. In addition, we are delighted to report feasibility of our novel voice-over digitalised aid to provide decision support for parents to vaccinate their children against influenza in a pilot study which has been published:

The decision support aid also provides the parents of both the benefits and the potential risks of vaccination against a common infectious disease among children. The content of the decision support aid can be extended to other vaccines in the National Childhood Immunization Programme.

Individual-vaccine plans are ideal but the local public primary health care system poses several challenges in its implementation, in contrast to a universal childhood vaccination schedule. Each public primary care clinic manages between 20-80 children during office hours every workday. To provide every child with individual vaccine plan will tax the limited healthcare manpower, potential difficulty in monitoring of vaccine efficacy at a population level due to variation in dosing and vaccination schedules. Nonetheless, for children with adverse events following immunisations (AEFI) and/or special needs, the nurses in the institution will review and personalize their vaccination plan accordingly in consultation with the primary care physician.

Reviewer 2 Report

After reading paper I have the following remarks.

1.     The article faces a very interesting topic and provides fascinating insight regarding the impact of vaccination policy changes in a developed country. The writing style is clear and easy to read and understand. I only have a few minor remarks.

2.     Line 48: the acronym OECD should be preceded by its full form.

3.     One of the main focuses of the article is parents’ vaccination hesitancy regarding their offspring. In particular, the changes in Singapore’s children vaccination schedule improved the aspect of convenience, as they diminished the mean number of vaccination sessions required. I advise the authors to briefly talk about this specific aspect of vaccination hesitancy and delay and focus in organizational and educational interventions to reduce it.

·      doi: 10.1155/2019/6764154

·      doi: 10.1097/INF.0000000000003499

4.     The safety question should be considered. Authors should evaluate literature data of combination vaccines’ safety and focus on possibility to implement passive or active surveillance program of adverse events following immunization

·      doi: 10.1080/21645515.2019.1704124

·      doi: 10.1080/21645515.2021.2021711

Author Response

Introduction

  1. Line 48: the acronym OECD should be preceded by its full form

Thank you for highlighting the amendment and it has been changed accordingly.

Discussion

  1. Discuss about convenience brought about by reduced vaccination sessions and focus on organizational/educational interventions that help reduce this hesitancy

We agree with the reviewer on the convenience due to the use of combination vaccines. We have included additional discussion points to highlight innovation which has been pilot-tested in the institution to support parental decision making in vaccinating their children against influenza and reduce their hesitancy. The outcomes, which have been published, are favourable and the next step is to adopt it for other childhood vaccines and implement the parental decision-support aids across the primary care clinics in the institution. Further research will be conducted to determine the effectiveness and implementation of these decision-support aids in addressing vaccine hesitancy using a randomized controlled trial and scaling up the use of these aid via implementation science respectively.

  1. Discuss safety of combination vaccines and focus on passive/active surveillance programs for adverse events

Healthcare providers are mandated to report all adverse events following immunisations (AEFI) to the health authority in Singapore. Parent can report any vaccine related reaction to the primary care nurses in the institution, and the latter will relay the information electronically to the health authority. The latter will collate and compute the prevalence of vaccine adverse events and publish the surveillance report annually. The authority will alert the primary care providers of abnormal surge in the incidences of any vaccine adverse events for immediate remedial measures to ensure safety of the vaccine recipients.

Round 2

Reviewer 1 Report

No comments any more.